# Intratumoral Platelets: Harmful or Incidental Bystanders of the Tumor Microenvironment?

**DOI:** 10.3390/cancers14092192

**Published:** 2022-04-27

**Authors:** Ophélie Le Chapelain, Benoît Ho-Tin-Noé

**Affiliations:** Laboratory of Vascular Translational Science, U1148 Institut National de la Santé et de la Recherche Médicale (INSERM), Université Paris Cité, F-75006 Paris, France; ophelie.le-chapelain@inserm.fr

**Keywords:** platelets, tumor microenvironment, solid tumors, angiogenesis, lymphangiogenesis, vascular integrity, tumor-associated platelets

## Abstract

**Simple Summary:**

The tumor microenvironment (TME) is the complex and heterogenous ecosystem of solid tumors known to influence their growth and their progression. Besides tumor cells, the TME comprises a variety of host-derived cell types, ranging from endothelial cells to fibroblasts and immune cells. Clinical and experimental data are converging to indicate that platelets, originally known for their fundamental hemostatic function, also participate in tumor development and shaping of the TME. Considering the abundance of antiplatelet drugs, understanding if and how platelets contribute to the TME may lead to new therapeutic tools for improved cancer prevention and treatments.

**Abstract:**

The tumor microenvironment (TME) has gained considerable interest because of its decisive impact on cancer progression, response to treatment, and disease recurrence. The TME can favor the proliferation, dissemination, and immune evasion of cancer cells. Likewise, there is accumulating evidence that intratumoral platelets could favor the development and aggressiveness of solid tumors, notably by influencing tumor cell phenotype and shaping the vascular and immune TME components. Yet, in contrast to other tumor-associated cell types like macrophages and fibroblasts, platelets are still often overlooked as components of the TME. This might be due, in part, to a deficit in investigating and reporting the presence of platelets in the TME and its relationships with cancer characteristics. This review summarizes available evidence from clinical and animal studies supporting the notion that tumor-associated platelets are not incidental bystanders but instead integral and active components of the TME. A particular emphasis is given to the description of intratumoral platelets, as well as to the functional consequences and possible mechanisms of intratumoral platelet accumulation.

## 1. Introduction

The local ecosystem of tumors or tumor microenvironment (TME) has gained considerable interest because of its decisive role in maintaining a supportive niche for cancer cells. Among its numerous effects, the TME can indeed favor the proliferation, dissemination, and immune evasion of cancer cells, and can therefore impact therapy and disease recurrence. Components of the TME are highly heterogeneous and vary greatly with the cancer location, type, and stage, and can also be affected by chemotherapy [1,2]. The TME comprises an extracellular matrix, matricellular proteins, cytokines, and growth factors, as well as many different cell types, ranging from cancer cells themselves to a variety of non-malignant cells. Non-malignant cells of the TME include both nearby endogenous cells normally present in the affected tissue, such as fibroblasts and adipocytes, and cells recruited from distant sites, including immune cells and mesenchymal stem cells. Recent studies indicate that the TME also contains non-dividing supportive cells arising directly from cancer stem cells [3,4]. Tumor-associated macrophages (TAM), cancer-associated fibroblasts (CAFs), and vascular endothelial cells, together with their supporting pericytes, have been arguably among the most prominent and intensively scrutinized stromal cells of the TME [5,6,7]. Lymphocytes and adipocytes have also had their rightful share of attention [2,5,7,8]. On the contrary, although they have long been recognized to support hematogenous metastatic dissemination [9], platelets have been largely omitted from the TME equation.

Platelets are the second most numerous of the circulating blood cells. Historically known as the primary effector cells of haemostasis and thrombosis due to their ability to aggregate and promote coagulation, platelets are also recognized for their pivotal role in other processes, such as innate and adaptive immune responses [10]. Platelets are anucleated cells arising from the cytoplasmic fragmentation of megakaryocytes whose diameter of ~2–3 μm is considerably smaller than those of other circulating blood cells. Despite their small size and lack of nucleus, platelets are actually highly versatile cells whose functions far exceed the formation of blood clots. It is now well established that platelets regulate leukocyte recruitment and activation [11,12] and participate in angiogenesis [13,14,15], lymphangiogenesis [16,17,18], and tissue remodeling in general, as in wound healing [16,19,20], which all contribute to TME formation. The ability of platelets to exert such a wide range of effects is due to the fact that, in addition to their multiple receptors and interaction partners, platelets release numerous bioactive molecules stored in their secretion granules [21]. The lifespan of platelets is approximately 10 days in humans and 5 days in mice. In view of the time required for cancer to develop and spread representing years, this lifespan appears incompatible with a lengthy presence of platelets at the individual level in the tumor stroma. However, the continuous supply and renewal of platelets at a daily rate of 10^11^ platelets a day [22] is likely sufficient to enable a persistent presence of platelets in tumors. In the present review, we summarize and discuss available evidence that tumor-associated platelets (TAP) exist and are not incidental bystanders but instead represent integral and active components of the TME.

## 2. Clinical Evidence That Platelets Impact Tumor Progression and Response to Therapy

Evidence that cancers can impact platelets came very early and even preceded the discovery that platelets mediate thrombus formation, with the observation by Armand Trousseau in 1865 that patients with late-stage cancer frequently developed thrombotic complications [23]. Since then, this observation has been largely confirmed, with thrombosis representing the second leading cause of death in cancer patients, just behind infections [24]. Moreover, and more importantly with respect to the focus of the current review, it has become clear that influences between platelets and cancers go in both directions. Various correlations between platelet parameters and cancer prognosis have indeed been reported. Thrombocytosis, which is defined as a platelet count greater than 400 × 10^3^ platelets/µL blood, has been found to be associated with shorter survival and poor prognosis in a variety of solid cancers, including lung [25], breast [26], kidney [27], glioblastoma [28], pancreatic [29], ovarian [30], and gastrointestinal [31] cancer. Remarkably, in all of these examples, the correlation of thrombocytosis with an adverse outcome was maintained in multivariate analyses, including adjustment for the clinical stage, thus making of an elevated platelet count an independent predictor of shorter survival. It is also important to note that a worsening of cancer prognosis in patients with thrombocytosis was not associated with an increased incidence of thromboembolism.

There is emerging evidence that, like platelet count, the plateletcrit (PCT, the volume occupied by platelets in blood) and the mean platelet volume (MPV) may also bear a prognostic value in some cancers. For instance, a more elevated preoperative plateletcrit was recently found to be an independent marker of poor prognosis in patients with resectable non-small cell lung cancer [32]. Regarding MPV, there are conflicting reports of either greater or lesser MPV being associated with cancer prognosis. While lower MPV was found to predict poor prognosis in renal cell carcinoma [33], pancreatic cancer [34], and bladder cancer [35], quite the opposite was found in colorectal [36] and gastric cancer [37], with elevated MPV instead being associated with decreased overall survival. These inconsistencies might be related to the type of cancer but also to the fact that MPV remains a controversial marker due to its lack of specificity. Indeed, variations in MPV can reflect different aspects of platelet biology, including platelet reactivity and activation status, but also age and turnover rate [38].

Several studies have indicated that, in addition to overall cancer progression and outcome, platelets can provide information on treatment response and recurrence risk. For instance, normalization of platelet counts following chemotherapy was found to be indicative of a good treatment response in patients with lung [39] or ovarian [40] cancer, while lack of platelet count normalization was associated with an increased risk of recurrence. Co-culture experiments of platelets with various human cancer cell lines have shown that by providing proliferative and survival signals, platelets can directly counteract the antiproliferative and cytotoxic action of several chemotherapeutic drugs, including gemcitabine [41], cisplatin [39], and docetaxel [30,40]. Thus, platelets may not just be markers of a response to chemotherapy but may directly impair it as well. Interestingly, results from recent studies suggest that platelets may participate in immune checkpoint interactions, whose inhibition has become a new standard adjuvant treatment for several cancers. Programmed death-ligand 1 (PD-L1), a prominent immune checkpoint protein expressed by antigen-presenting cells and by a variety of tumor cells, was detected on platelets of patients with PD-L1-positive tumors, but not on platelets from healthy individuals or from patients with PD-L1-negative tumors [42,43,44]. PD-L1-expressing platelets were found both in the circulation and within tumors, and possessed the ability to inhibit CD4 and CD8 T-cells [44], suggesting that they might favor PD-L1-mediated tumor immune evasion. Because levels of platelet PD-L1 were found to mirror PD-L1 expression in tumors [44], it was proposed that PD-L1 on platelets could help to identify patients who would benefit from inhibitors of the PD-1/PD-L1 axis, the main target of current checkpoint therapies. In support of this notion, high levels of platelet PD-L1 were found to predict a good response to anti-PD-1 antibodies in patients with PD-L1-positive lung cancer [44]. Besides its potential prognostic value, the expression of PD-L1 on platelets raises several questions. First, could anti-PD-L1 therapy cause thrombocytopenia or thrombosis via elimination or activation of PD-L1-positive platelets? With reports of up to more than 50% PD-L1-positive platelets in patients with head and neck squamous cell carcinoma or lung cancer [43,44], these issues may warrant attention when assessing the adverse effects of anti-immune checkpoint therapies. Another concern is whether PD-L1 on platelets could act as a decoy for binding of anti-PD-L1 therapeutic antibodies intended to target cancer cells.

Another line of evidence pointing to a contribution of platelets to tumor development and progression comes from the anticancer effects of antiplatelet drugs. Daily low dose acetylsalicylic acid (aspirin), an irreversible inhibitor of cyclooxygenases (COX-1 and COX-2), has been recommended since 2007 for primary prevention of colorectal cancer by the US Preventive Services Task Force [45]. Moreover, several studies suggest that another antiplatelet drug, clopidogrel, an antagonist of the purinergic P2Y12 receptor, also reduces the incidence of colorectal cancer [46]. Nevertheless, COX and P2Y12 are not specific to platelets and can also be expressed by other cell types, including various cancer cell types [47,48,49,50,51,52]. Therefore, one cannot exclude that part of the anticancer effects of antiplatelet drugs may be independent of platelets. However, there are strong arguments (reviewed in [53,54]) indicating that aspirin and clopidogrel do exert their anticancer effects through platelet inhibition. In particular, with respect to aspirin, its short half-life (20 min), combined with the fact that its anticancer effects have been observed at a low-dose sufficient to irreversibly and completely inhibit the activity of COX-1 in platelets but not that of renewable COX-2 in cancer cells or other nucleated cells, make a significant contribution due to non-platelet cells being unlikely.

## 3. Tumor Cell-Platelet Interactions: Mechanisms and Functional Consequences

The clinical evidence that platelets play a role in the thrombotic complications and progression of cancers has stimulated intensive research on the interactions and mechanisms that could underlie these effects. It is now well established that platelets and tumor cells are able to interact through paracrine and direct physical contacts [55]. Various ligand/receptor couples have been reported to mediate these contacts. For example, ADAM9 was identified as a counter receptor for platelet integrin α6β1 on two different types of mouse tumor cells [56], and galectin-3 and podoplanin as counter-receptors for platelet glycoprotein VI (GPVI) [57] and CLEC-2 [58], respectively. Cadherin-6, a marker and mediator of epithelial-mesenchymal transition (EMT), is also expressed by platelets [59] and was recently shown to support contacts between platelets and tumor cells via homophilic and heterotypic interactions [60]. Other platelet receptors such as GPIIb/IIIa [55], α2β1 integrin [61], and P-Selectin [62] have also been shown to support direct platelet-tumor cell interactions, but their exact ligands on tumor cells have not been identified to date.

Platelet-tumor cell interactions have multiple and bidirectional functional consequences. With respect to platelet function, these interactions have long been known to cause platelet activation and aggregation [63,64], an effect with obvious relevance not only to the adhesion and dissemination potential of cancer cells, but also to the risk of cancer-associated thrombosis initially described by Trousseau [23]. Nevertheless, it is worth noting that tumor-induced platelet responses do not univocally ultimately result in platelet aggregation. Recent data by Plantureux et al. indeed indicate that contacts between tumor cells and individual platelets can trigger microvesicle production by platelets without causing aggregation [60]. Likewise, in a much earlier electron microscopy study depicting the attachment of tumor cells to circulating platelets and vessel walls in vivo, Warren et al. reported that “*the contact between platelets and tumor cells did not elicit a wave of platelet-platelet adhesion such as occurs when a white thrombus is formed*” [65].

Many different non-exclusive pathways have been reported to participate in cancer-induced platelet activation. These pathways include both cell contact-dependent and -independent mechanisms. Excellent comprehensive reviews on this matter can be found elsewhere [66,67] and thus just a brief overview will be given here. Cancer-induced platelet activation can occur via the generation of small amounts of thrombin by tumor cells- or tumor microvesicle-expressed tissue factor [68,69,70], the secretion of ADP [68,70] or cathepsin B [71] by tumor cells, tumor cell-induced generation of neutrophil extracellular traps (NETs) [72,73], tumor cell-induced release of ADP and thromboxane A2 by platelets [63,70], or via direct engagement of platelet receptors with signaling properties such as CLEC-2 [58] and cadherin 6 [60] by tumor cell ligands.

As indicated by clinical observations, in addition to platelet activation, cancers can also enhance thrombopoiesis and influence platelet phenotype. To date, cancers have been shown to promote thrombopoiesis by at least 3 mechanisms: the production by tumor cells of interleukin 6 [74,75], that of granulocyte- and granulocyte-macrophage colony stimulating factors [76], and, possibly, of thrombopoietin [77,78]. One mechanism by which modifications of platelet phenotype occurs in cancers is the active and selective sequestration of cancer-derived secreted proteins such as VEGF-A and other growth factors by circulating platelets [14,79]. Transfer of proteins from cancer cells to platelets can also occur during transient direct cell-cell contacts, as was recently shown for the acquisition of functional PD-L1 by platelets from PD-L1-expressing tumor cells [44]. In addition to protein content, cancers have been shown to impact platelet mRNA content. Cancers can transfer mRNA to platelets through the emission of microvesicles by tumor cells [80,81] and might also alter the platelet transcriptome through modulation of platelet pre-mRNA splicing [81,82]. Interestingly, cancer-induced phenotypic alterations of platelets are not solely considered across the spectrum of their possible functional consequences, but also with respect to their prognostic potential as biomarkers of the presence of cancer or its progression [43,44,80,83].

While cancer can alter thrombopoiesis quantitatively and modify the cargo of mature circulating platelets, there is evidence that cancer can also alter the platelet transcriptome and proteome by acting at the megakaryocyte level. For instance, in 2010, Zaslavsky et al. showed that thrombospondin-1 mRNA levels were up-regulated in megakaryocytes of tumor-bearing mice [84]. Interestingly, thrombospondin-1 has emerged as a potential regulator of the TME that would play a role in tumor vessel growth and function, as well as in escape from innate and adaptive antitumor immunity [85]. Its upregulation in platelets in cancer thus resonates particularly well with a platelet contribution to the shaping of the TME. As discussed further below, there is mounting evidence that, like inflammation [86], solid cancers might trigger megakaryopoiesis programs distinct from the traditional ones.

With respect to cancer cells, the most reported and scrutinized consequence of their interactions with platelets is arguably the increased metastatic potential these interactions induce. Since the seminal study by Gasic et al. [9], numerous experimental studies in rats and mice using a variety of tumor cell lines have confirmed the ability of platelets to promote tumor foci formation in the lungs [55,56,57,58,62,87,88,89] and have extended this observation to other organs such as the liver [62,88,90] and kidneys [62]. Because the pro-metastatic potential of platelets was mostly demonstrated using models of hematogenous metastases relying on the injection of tumor cells directly into the bloodstream, it is noteworthy that it was also observed in models of spontaneous metastasis [56,57,89,90]. Platelets interacting with cancer cells promote their metastatic potential by stimulating their epithelial-mesenchymal transition (EMT) [60,61,91,92], by protecting them from NK cell antitumor immunity [93,94], and by enhancing their adhesiveness to the vessel wall and subsequent extravasation [57,65,95]. Apart from enhancing the metastatic potential of cancer cells, platelets can also influence cancer cell survival and proliferation via direct interactions and paracrine signaling. Platelets have been shown to promote the proliferation of various human and mouse cancer cell lines in vitro [96,97,98,99,100,101], an effect attributed in part to the release of growth factors such as transforming growth factor-β (TGF-β) by activated platelets [96,98,101]. Not all studies agree on the pro-proliferation effect of platelets on cancer cells. Several studies have described that, on the contrary, platelets can exert anti-proliferative and even cytotoxic effects on cancer cells in vitro [100,102,103,104,105]. These discrepancies regarding the impact of platelets on cancer cell proliferation and survival in vitro also apply to in vivo studies, as platelets were found to have no impact [87,99,106,107], a positive impact [74,96,108,109,110,111], or a negative [60,105,112] impact on primary tumor growth in mouse models of solid cancers.

Collectively, these data show that a large number of mechanisms can support platelet-cancer cell interactions, and that the resulting biological effects can vary greatly. In this context, it is worth stressing that the type of cancer cell [89,113,114], as well as the localization of platelet-cancer cell interactions (i.e., in the bloodstream or locally at the primary tumor site) [60], have both been identified as important determinants of the mechanisms and functional consequences of platelet-cancer cell interactions.

## 4. Intratumoral Platelets: Occurrence and Possible Origins

Intriguingly, although many studies have highlighted the possible mechanisms and consequences of direct contacts and paracrine communication between platelets and cancer cells, it remains unclear if and how such interactions occur within primary tumors and their microenvironment. There is little if not no doubt that platelets and cancer cells interact closely together once cancer cells have entered the bloodstream, with the formation of tumor cell/platelet aggregates having been detected in models of experimental metastasis [65,115,116]. The lack of reports of such interactions in cancer patients can be easily explained by the scarcity of circulating tumor cells and the technical difficulties involved in detecting and isolating them [117].

In contrast to interactions in the bloodstream, evidence of interactions between platelets and cancer cells at the primary tumor site is scarcer. Yet, over the last 10 years, several experimental and clinical studies have provided data on tumor-infiltrating platelets (a list of studies describing intratumor platelets is given in Table 1).

In 2012, Stone et al. reported the presence of platelets in the tumor perivascular and extravascular compartments in a mouse model of ovarian cancer [74]. Further observations of intravascular, perivascular, and extravascular platelets within tumors have since been made in mouse models of colorectal and brain cancer [60,110,130]. In humans, intravascular and extravascular platelets have also been found in the stroma of various types of solid tumors, including breast, lung, pancreatic, gastric, and colorectal adenocarcinoma [118,119,122,127,128,129] (Figure 1).

Although the studies and information on tumor-infiltrating platelets and their impact on and relation to cancer progression in humans are still limited, currently available data suggest that the presence of platelets in the tumor stroma is unlikely to be incidental. In fact, tumor-infiltrating platelets have been shown to be associated with the advanced stage of colorectal cancer [127], expression of EMT markers in pancreatic, and breast cancer [118,128], as wells as with chemoresistance and poor overall survival in breast, gastric, and pancreatic cancer [119,120,122,128]. These data, together with the observation that platelets are preferentially localized at the invasive front of pancreatic and breast tumors [118,128], suggest that, like cancer-associated thrombocytosis, the occurrence of tumor-infiltrating platelets may be indicative of an aggressive cancer phenotype.

How do platelets end up in the extravascular tumor microenvironment? One mechanism by which platelets can reach the extravascular space and interact directly with cancer cells is through the occurrence of intratumoral bleeding. Indeed, spontaneous intratumoral bleeding, related to tumor angiogenesis or to tumoral invasion, occurs in a variety of cancers [132,133,134,135,136]. Whether sporadic intratumoral bleeding events are sufficient to ensure a continuous presence of platelets in the tumor stroma is, however, uncertain. Nonetheless, even if they are spatially scattered and episodic during the course of cancer progression, platelet-cancer cell interactions subsequent to intratumoral bleeding may be sufficient to cause consequential phenotypic changes in cancer cells.

Reports of extravasated platelets found in the absence of bleeding in various inflamed organs [137], including experimental tumors [60,109,124,130], suggest that platelets may also access the extravascular tumor stroma via transmigration, either directly, or through the association with transmigrating leukocytes. The fact that platelet-specific deficiency in focal adhesion kinase [109], a protein known for its role in cell adhesion and migration, or deficiency in P-Selectin [124], a protein central to platelet-leukocyte interactions, is associated with reduced platelet deposition within the microenvironment of experimental tumors argues in favor of this possibility. However, it should be noted that, although the ability of platelets to migrate within the intravascular compartment was recently demonstrated [138,139], direct observation of active transmigration or of platelets migrating in the extravascular space has yet to be provided.

It has become clear over the last years that adaptative megakaryopoiesis programs can be triggered in inflammatory conditions [86,140]. In addition, extramedullary hematopoiesis has been shown to occur in various solid tumors [131,141,142,143,144]. Therefore, apart from blood-borne platelets, a subset of intratumoral platelets may also originate from local production programs, as suggested by the detection of megakaryocytes within the tumor stroma of patients with brain, hepatic, renal, or breast cancer [131,142,143,144] (Figure 1).

Along with platelets infiltrating the extravascular tumor stroma, platelets interacting with the tumor vasculature likely account for a substantial fraction of intratumoral platelets. Indeed, inflammation and angiogenesis are constitutive features of solid cancers, and platelets are now known to continuously interact with and accumulate in blood vessels at sites of inflammation [138,145] and angiogenesis [13]. Furthermore, platelet accumulation in the tumor vasculature can also occur through intratumoral thrombosis [58,146,147]. Finally, circulating platelets may also interact directly with cancer cells at sites of vascular mimicry, which corresponds to areas where cancer cells organize themselves into vascular channels to supply blood independently of endothelial cells [148].

## 5. Shaping of the Tumor Microenvironment by Platelets

### 5.1. Platelets and Tumor Angiogenesis and Vascular Integrity

Studies in animal models of solid tumors strongly indicate that the functional relevance of intratumoral platelets exceeds the regulation of cancer cell phenotype, proliferation, or survival. There is converging data in favor of a role of intratumoral platelets in shaping the TME, in particular its vascular compartment. Platelets are well-known for containing a variety of angiogenic factors in their alpha granules and have been shown to participate in angiogenesis in a variety of experimental settings, including models of solid tumors [13,74,107,149,150]. The involvement of platelets in tumor angiogenesis ranges from stimulating the proliferation of endothelial cells [99,150], promoting the recruitment of pericytes [74,88,107] and that of bone marrow-derived cells [149], to maintaining tumor vessel function and integrity [88,107,108,151,152]. As a result of these activities, depletion of platelets or targeting of their activation receptors has been shown to result in reduced tumor vessel density [74,107,149], maturation [74,88,107], and functionality [58,88,107,108,151,152]. Thus, experimental studies indicate that platelets regulate tumor angiogenesis not only quantitatively but also qualitatively, notably by continuously preventing tumor vessel leakage and bleeding [88,107,108,151,152]. The latter supportive functions of platelets towards tumor vessel integrity emphasize the importance of the physical presence of platelets within tumors for regulating the TME. Indeed, the stabilization of both angiogenic and inflamed vessels by platelets requires direct contacts between platelets and such vessels [13,145]. Interestingly, several studies in mouse models of solid cancers are converging to suggest that targeting the vasculoprotective function of platelets in tumors can enhance the intratumor delivery and antitumor effects of chemotherapeutic drugs such as paclitaxel via the induction of tumor vascular leakiness [106,108,153].

Despite evidence from experimental models, if and how tumor platelet content correlates with tumor angiogenesis in cancer patients has not been investigated. Additionally, it is worth mentioning that platelet depletion had no impact on tumor vessel density in a mouse model of glioblastoma [99], which suggests that the contribution of platelets to tumor angiogenesis may vary with the cancer type.

### 5.2. Platelets and Tumor Lymphangiogenesis

Platelets are now well-established actors of lymphangiogenesis. They ensure proper separation of blood and lymphatic vessels during development, notably by regulating the proliferation, migration, and tube formation of lymphatic endothelial cells (LECs) through engagement of their CLEC-2 receptor by podoplanin on LECs [17,125]. Platelet CLEC-2 is also required for the development and maintenance of lymph nodes [154]. Remarkably, platelets are not required for maintaining the mature blood and lymphatic systems separated in the absence of challenges post-development [155]. However, recent studies in mice have indicated that, upon vascular remodeling, such as during wound-healing or in the TME, platelets again intervene in lymphangiogenesis by stimulating lymphatic growth via the secretion of VEGF-C [16] and by preventing the mixing of lymphatic and blood circulations through the engagement of CLEC-2 [155]. Considering that lymphatics provide routes for the dissemination of cancer cells, these data suggest that platelets may promote metastasis not only via their direct interactions with cancer cells but also through shaping the TME and giving cancer cells access to regional lymph nodes. The description of a positive correlation between intratumoral platelet content and both lymphatic vessel density and lymphovascular invasion in human esophageal cancer supports this hypothesis [125].

### 5.3. Platelets and the Tumor Immune Microenvironment

In addition to their role in tumor blood and lymphatic network formation, platelets may potentially also participate in setting up the immune component of the TME. Platelets indeed regulate the infiltration and functions of leukocytes in many inflammatory diseases and conditions [12,156]. Furthermore, platelets were found to recruit neutrophils to form early metastatic niches in the lungs of mice injected intravenously with various types of tumor cells [157]. Intriguingly, however, there are very few reports linking platelets to tumor immune cell content. Among the sparse available data, induction of acute thrombocytopenia was shown to have no significant impact on neutrophil and macrophage infiltration into the stroma of subcutaneously implanted Lewis lung carcinoma tumors in mice [152]. In contrast to these results, there is evidence supporting the role of platelets in controlling intratumor T cell infiltration. As mentioned above, functional PD-L1 was found on platelets of patients with PD-L1-positive lung cancers [44]. Interestingly, high levels of PD-L1 on platelets were associated with lower numbers of infiltrating T cells in the TME [44]. In agreement with these results, in mice, platelet depletion caused an increase in T cells in the TME of a colon adenocarcinoma model, an effect that could be reverted by transfusion of PD-L1-positive but not PD-L1-negative platelets [111]. Thus, together, these clinical and experimental results suggest a possible role of platelet PD-L1 in tumor immune evasion [111].

Finally, the observation of platelets interacting with podoplanin-expressing CAF in pancreatic cancer [121] or in peritoneal metastasis of gastric cancer [123], suggests that platelets may influence the activities of other major actors of the TME.

## 6. Conclusions

Despite accumulating evidence that platelets can impact the development and aggressiveness of solid tumors, a more systematic assessment of their presence in the TME and of how it relates to cancer characteristics and other TME components appears necessary. Such investigations could help to improve our fundamental understanding of the platelet-cancer crosstalk and enable the development of new diagnostic, prognostic, and therapeutic tools for solid cancers. In addition, they should allow a better appreciation of the cancer type-specific aspects of the platelet actions towards solid tumors, as hinted by earlier reports on mouse models. Yet, despite the current blind spots in our knowledge of intratumoral platelets, they should now be recognized as integral components of the TME, at least for their contribution to tumor vasculature formation and stabilization.

## Figures and Tables

**Figure 1 cancers-14-02192-f001:**
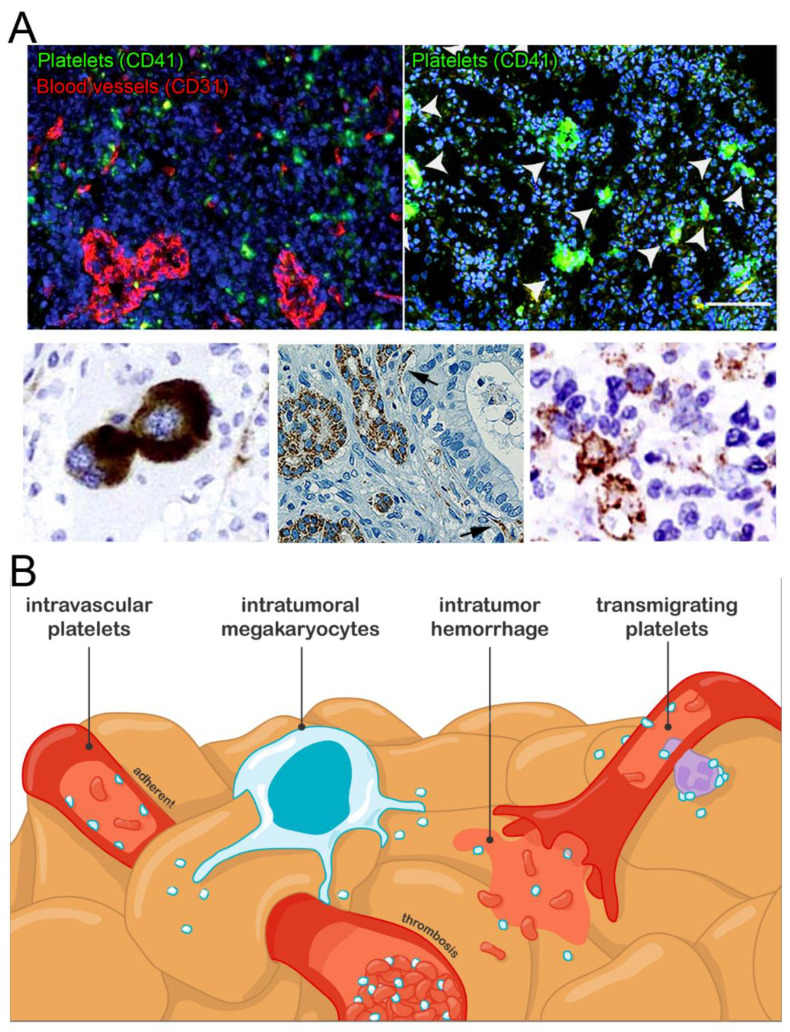
Intratumoral platelets: occurrence and possible origins. (**A**). Images of the different types of intratumoral platelet localization and interactions reported: perivascular aggregates (upper left panel, mouse ovarian cancer), aggregates with tumor cells (upper right panel: mouse glioma, platelet aggregates are indicated by white arrows, scale bar = 100 µm; bottom right panel: colorectal cancer), intravascular platelets (black arrows, lower middle panel: human pancreatic cancer), and intratumor megakaryocytes (lower left panel: human renal cell carcinoma). Adapted with permission from Refs. [109,119,126,130,131]. Copyright 2016, American Society for Clinical Investigation; Copyright 2018, Society of Surgical Oncology; Copyright 2021, UICC; Copyright 2019, American Society of Hematology; Copyright 2014, Elsevier. (**B**). Schematic representation of the possible mechanisms supporting intratumoral platelets accumulation.

**Table 1 cancers-14-02192-t001:** Clinical and preclinical reports of intratumoral platelet occurrence.

Tumor Type	Platelet Marker	Involved Pathway	Main Observations Pertaining to the Presence of Intratumoral Platelets	Refs.
Pancreatic cancer	CD42b	ND	Extravascular platelets surrounding tumor cells were detected at the tumor invasive front, in association with the expression of epithelial mesenchymal transition markers	[118]
			The presence of intratumoral extravascular and intravascular platelets was associated with poor post-surgical survival and recurrence-free survival	[119,120]
			Extravascular platelets were found around cancer-associated fibroblasts in the tumor stroma. Chemotherapy led to a decrease in both cancer-associated fibroblasts and intratumoral platelets.	[121]
Gastric cancer	CD42b	ND	Extravascular platelets were found around cancer-associated fibroblasts. The presence of intratumoral platelets was associated with chemoresistance and poor overall survival.	[122]
			The presence of extravascular platelets around tumor cells and cancer-associated fibroblasts in peritoneal metastatic lesions was associated with poor overall survival.	[123]
	CD41		Extravascular platelets were found accumulating around tumor cells.	[124]
Esophageal cancer	CD61	ND	The presence of extravascular platelets was associated with shorter disease-free survival and increased lymphangiogenesis and lymphovascular invasion.	[125]
Colorectal cancer	CD41 CD42b	ND	Intratumoral platelet content increased with tumor stage and lymph node metastasis. Extravascular platelets were found accumulating around tumor cells and tumor vessels. Platelet infiltration is associated with a poor prognosis in postsurgical colorectal cancer patients	[60,124,126,127]
Breast cancer	CD42b	ND	Extravascular and perivascular platelets were detected at the tumor invasive front, in association with the expression of epithelial mesenchymal transition markers, and their presence was associated with chemoresistance.	[128]
	CD41	ND	Extravascular platelets were found accumulating around tumor cells.	[124,129]
Bowel cancer	CD41	ND	Extravascular platelets were found accumulating around tumor cells.	[129]
Lung cancer	CD41	ND	Extravascular platelets were found accumulating around tumor cells.	[129]
			Increased amount of intratumoral platelets in patients with a positive response to PD-L1 therapy.	[111]
			Extravascular platelets were found in PD-L1-negative and -positive lung cancer.	[44]
Hepatocellular cancer	CD41	ND	Extravascular platelets were found accumulating around tumor cells.	[124]
Ovarian cancer (mice)	CD42b	ND	Perivascular and extravascular platelets were found in abundance in the tumor stroma.	[74]
		FAK	Platelet-specific deficiency in FAK reduced intratumoral platelet content and tumor growth.	[109]
		TGFβ	Platelet-specific deficiency in TGFβ partly reduced intratumoral platelet content.	[110]
		Gα13/ Gi	Platelet-specific deficiency in Gα13 or Gi protein reduced intratumoral platelet content and tumor growth.	[96]
Insulinoma and melanoma (mice)	CD42b	P selectin	P-selectin deficiency reduced intratumoral platelet content and tumor growth.	[124]
Intestinal cancer (mice)	CD41	P selectin	P-selectin deficiency reduced intratumoral platelet content and tumor growth.	[127]
Colorectal cancer (mice)	CD42b	ND	Perivascular and extravascular platelets were found in the tumor stroma, preferentially at the tumor periphery.	[60]
Glioma (mice)	CD41	Podoplanin	Podoplanin expressed by cancer cells is required for intratumoral platelet infiltration	[130]
Breast cancer, fibrosarcoma, Burkitt’s lymphoma (mice)	CD41	ND	Extravascular platelets were found in abundance in the tumor stroma.	[129]

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
