# Peer review of "Intratumoral Platelets: Harmful or Incidental Bystanders of the Tumor Microenvironment?"

_cancers, 2022, doi:10.3390/cancers14092192_

Round 1

Reviewer 1 Report

The aim of this manuscript is to explore available evidence, from clinical and animal studies, supporting the notion that tumor-associated platelets are active components of the tumor microenvironment (TME). To this aim, authors provide a detailed description of intratumoral platelets and the functional consequences of their accumulation.

This manuscript shows rich content, providing a deep insight for some works: I found it to be well-written and accessible, providing sufficient information for the non-expert while also achieving a balance of detail for those with more expertise in the field. This is the additional point, which makes this manuscript original, in comparison to published literature. Even if the manuscript provides an organic overview, with a densely organized structure and based on well-synthetized evidence, there are aspects to be mentioned, to make the article fully readable. For these reasons, the manuscript requires minor changes.

Please find below an enumerated list of comments on my review of the manuscript:

INTRODUCTION:

LINE 40: Platelets represent a cellular subgroup of the elements, circulating in the bloodstream, with multiple and pivotal role, in several processes, from primary hemostasis to innate immunity, as suggested by several and recent studies (see, for reference: Bianchi, S.; Torge, D.; Rinaldi, F.; Piattelli, M.; Bernardi, S.; Varvara, G. Platelets’ Role in Dentistry: From Oral Pathology to Regenerative Potential. Biomedicines 2022, 10, 218. https://doi.org/10.3390/biomedicines10020218), that examinated their multiple biological functions. In this context, the manuscript will benefit from providing an organic and, at the same time, accessible introduction to platelets, providing recent evidence for the non - expert in the field. This is the minor point of this introductive section.

LINE 186: Thrombospondin – 1 may exert multiple and pivotal role, in the tumor microenviroment, due to its interaction with signaling receptors, angiogenic and imunne-modulatory factors, as suggested by several and recent studies (see, for reference: Kaur, S.; Bronson, S.M.; Pal-Nath, D.; Miller, T.W.; Soto-Pantoja, D.R.; Roberts, D.D. Functions of Thrombospondin-1 in the Tumor Microenvironment. Int. J. Mol. Sci. 202122, 4570. https://doi.org/10.3390/ijms22094570), which evaluate its molecular function, in the tumor microenvironment. For these reasons, the authors should remark this aspect, in order to complete the information and provide to the readers recent evidence on this topic.

As regards the main topic, it is interesting and certainly of great scientific and clinical impact: in fact, this review touches a significant area, by analyzing the available evidence on potential role of platelets in tumor microenvironment (TME). As regards the originality and strenghts of this manuscript, this is a significant contribute to the ongoing research on this topic. Overall, the contents are rich, and the authors also give their deep insight for some works.

There is a specific and detalied explanation for the majority of evidence used in this study: this is particularly significant, since the manuscript relies on a multitude of cellular and molecular analysis, to derive its conclusions.

The conclusion of this manuscript is perfectly in line with the main purpouse of the paper: the authors have designed and conducted the study properly. As regards the conclusions, they are well written and present an adequate balance between the description of previous findings and the results presented by the authors.

Finally, this manuscript also presents a basic structure, properly divided and characterized by organic and detailed figures and tables. This manuscript looks like very informative since there are few evidence on this topic. As regards tables and figures, they are legible and easy to follow.

In conclusion, this manuscript is densely presented and well organized, based on well-synthetized evidences. The authors were lucid in their style of writing, making it easy to read and understand the message, portrayed in the manuscript. Besides, the methodology design was rigorous and appropriately implemented within the study. However, many of the topics are very concisely covered. This manuscript provided a comprehensive analysis of current knowledge in this field. Moreover, this research have futuristic importance and could be potential for future research. However, the minor concern of this manuscript is with the introductive section: for these reasons, I have minor comments only for the introductive section, for improvement before acceptance for publication. The article is accurate and provides relevant information on the topic and I suggest minor changes to be made in order to maximize its scientific impact. I would accept this manuscript, if the comments are addressed properly.

Author Response

We would like to thank both Reviewer’s and the Editor for giving us the opportunity to revise our manuscript and improve it with the help of your valuable suggestions and comments.

Reviewer 1.

INTRODUCTION: LINE 40: Platelets represent a cellular subgroup of the elements, circulating in the bloodstream, with multiple and pivotal role, in several processes, from primary hemostasis to innate immunity, as suggested by several and recent studies (see, for reference: Bianchi, S.; Torge, D.; Rinaldi, F.; Piattelli, M.; Bernardi, S.; Varvara, G. Platelets’ Role in Dentistry: From Oral Pathology to Regenerative Potential. Biomedicines 2022, 10, 218. https://doi.org/10.3390/biomedicines10020218), that examinated their multiple biological functions. In this context, the manuscript will benefit from providing an organic and, at the same time, accessible introduction to platelets, providing recent evidence for the non - expert in the field. This is the minor point of this introductive section.

We thank you for this suggestion. We have added a short sentence and reference to complete the introductory part on platelets.

LINE 186: Thrombospondin – 1 may exert multiple and pivotal role, in the tumor microenviroment, due to its interaction with signaling receptors, angiogenic and imunne-modulatory factors, as suggested by several and recent studies (see, for reference: Kaur, S.; Bronson, S.M.; Pal-Nath, D.; Miller, T.W.; Soto-Pantoja, D.R.; Roberts, D.D. Functions of Thrombospondin-1 in the Tumor Microenvironment. Int. J. Mol. Sci. 202122, 4570. https://doi.org/10.3390/ijms22094570), which evaluate its molecular function, in the tumor microenvironment. For these reasons, the authors should remark this aspect, in order to complete the information and provide to the readers recent evidence on this topic.

We thank you for pointing this out. We have added a sentence and a reference to stress the relevance of cancer-induced thrombospondin-1 up-regulation in platelets with respect to the TME :

« ….Interestingly, thrombospondin-1 has emerged as a potential regulator of the TME that would play a role in tumor vessel growth and function, as well as in escape from innate and adaptive antitumor immunity (Kaur, S et al. Int. J. Mol. Sci. 2021). Its upregulation in platelets in cancer thus resonates particularly well with a platelet contribution to shaping of the TME. »

Reviewer 2 Report

In this manuscript, Chapelain and Ho-Yin-Noé wrote a comprehensive review on platelets in cancer. The authors provided important references available in the literature. One issue that needs more clarification and emphasis is the impact of aspirin on primary and secondary cancer prevention. The authors briefly stated that the effect of aspirin in cancer might be related to its anti-platelet or direct effects on cancer cells. However, aspirin's short half-life and low-dose aspirin's anti-cancer effect make a significant contribution due to non-platelet cells unlikely; i.e., renewable COX-2 in cancer cells or other cells can not be effectively inhibited by a low daily dose of aspirin. Therefore, the main target of low-dose daily aspirin is COX-1 in platelets, and its anti-cancer effect is because of inhibiting platelets.   

Author Response

We would like to thank both Reviewer’s and the Editor for giving us the opportunity to revise our manuscript and improve it with the help of your valuable suggestions and comments.

Reviewer 2.

In this manuscript, Chapelain and Ho-Yin-Noé wrote a comprehensive review on platelets in cancer. The authors provided important references available in the literature. One issue that needs more clarification and emphasis is the impact of aspirin on primary and secondary cancer prevention. The authors briefly stated that the effect of aspirin in cancer might be related to its anti-platelet or direct effects on cancer cells. However, aspirin's short half-life and low-dose aspirin's anti-cancer effect make a significant contribution due to non-platelet cells unlikely; i.e., renewable COX-2 in cancer cells or other cells can not be effectively inhibited by a low daily dose of aspirin. Therefore, the main target of low-dose daily aspirin is COX-1 in platelets, and its anti-cancer effect is because of inhibiting platelets.   

We thank you for this suggestion, which we have taken into consideration. We have modified our text to incorporate these notions. The revised text now states :

« However, there are strong arguments (reviewed in [52,53]) indicating that aspirin and clopidogrel do exert their anticancer effects through platelet inhibition. In particular, with respect to aspirin, its short half-life (20 min) combined with the fact that its anticancer effects have been observed at a low-dose sufficient to irreversibly and completely inhibit the activity of COX-1 in platelets but not that of renewable COX-2 in cancer cells or other nucleated cells, make a significant contribution due to non-platelet cells unlikely. »
